# Cattle Cleanliness from the View of Swedish Farmers and Official Animal Welfare Inspectors

**DOI:** 10.3390/ani11040945

**Published:** 2021-03-27

**Authors:** Frida Lundmark Hedman, Maria Andersson, Vanja Kinch, Amelie Lindholm, Angelica Nordqvist, Rebecka Westin

**Affiliations:** 1Department of Animal Environment and Health, Swedish University of Agricultural Sciences, P.O. Box 234, 53223 Skara, Sweden; maria.andersson@slu.se (M.A.); rebecka.westin@slu.se (R.W.); 2The County Administrative Board in Halland, Slottsgatan 2, 30186 Halmstad, Sweden; vanja.kinch@lansstyrelsen.se; 3The County Administrative Board in Kalmar, Regeringsgatan 1, 39186 Kalmar, Sweden; amelie.lindholm@lansstyrelsen.se; 4The County Administrative Board in Västra Götaland, Klostergatan 13, 53230 Skara, Sweden; angelica.nordqvist@lansstyrelsen.se

**Keywords:** assessment, attitudes, herd level, law, clean cattle, management routines

## Abstract

**Simple Summary:**

Cleanliness is important for the health and welfare of cattle, but also for farm profitability, as dirtiness increases the risk of sick animals and can compromise milk and meat production. Swedish legislation states that all animals must be ‘clean enough’, but dirty cattle are commonly recorded in official inspections in Sweden. This study investigated the reasons for dirtiness and how inspectors handle cases of dirty cattle. Of the 371 cattle farms inspected, 49% had dirty cattle. However, the inspectors did not categorize all farms with dirty cattle as non-compliant, mainly using the argument that only a few animals were dirty. Therefore, in addition to knowing what characterizes ‘clean enough’ cattle, both inspectors and farmers need better guidance on when a farm is compliant, or non-compliant, with animal welfare legislation. Dirtiness in cattle was found to depend mainly on management routines on the farm, which is a promising result since routines can be improved.

**Abstract:**

Dirty cattle have been commonly recorded in official animal welfare inspections in Sweden for years. The relevant authorities have initiated work to better understand the causes of dirty cattle, in order to improve compliance and standardize the grounds for categorizing a farm as non-compliant with welfare legislation when dirty animals are present. This study investigated the occurrence of dirty cattle in official animal welfare controls, on Swedish cattle farms, and examined farmers’ views on the reasons for non-compliance and on key factors in keeping animals clean. The data used were collected by animal welfare inspectors at the county level during the regular official inspections of 371 dairy and beef cattle farms over two weeks in winter 2020. In addition to completing the usual inspection protocol, the inspectors asked farmers a set of questions relating to why their animals were clean or dirty. Dirty cattle were found on 49% of the farms inspected, but only 33% of the farms were categorized as being non-compliant with Swedish welfare legislation. According to inspectors and farmers, dirtiness in cattle depends mainly on management routines, which is a promising result since routines can be improved. The results also revealed a need for better guidance for inspectors and farmers on when dirtiness should be categorized as non-compliance with animal welfare legislation.

## 1. Introduction

Dirty cattle are commonly recorded by animal welfare inspectors in Sweden during official inspections [1]. Inspectors have found dirty animals during 20–25% of cattle inspections every year since at least 2013, without no decreasing trend [1]. When analyzing inspection reports, Keeling [2] found dirty animals to be one of the most common forms of non-compliance with animal welfare legislation in official inspections on Swedish farms, while Lundmark Hedman and co-workers [3] found dirty cattle or dirty lying areas to be the most common form of non-compliance in official inspections and in private inspections by the dairy company Arla. A high incidence of dirty cattle has also been reported in other countries, for example, Finland [4], Norway [5], Brazil [6], Spain [7], and India [8]. Within the European Union (EU), there is no species-specific animal welfare legislation for cattle, except for calves. However, some member states, such as Sweden, include all cattle in their national legislation. Both EU and national animal welfare legislation are typically mainly resource- and management-based, i.e., consisting of requirements related to risk factors in the animal husbandry system, interior design, and management procedures [9,10]. However, Sweden has introduced a few animal-based requirements, i.e., requirements focusing on the appearance of the animal [11]. For example, Swedish national regulations for cattle farming state that ‘animals shall be kept clean enough’ [12]. 

Cleanliness is important for the animal, but also for the farmer. Dirty cattle can be expensive for the farm, since dirtiness clearly increases the risk of developing health problems, and most likely impairs milk yield and meat production. Cow cleanliness is essential for ensuring hygienic milk production and the good welfare of dairy cows [5]. Cow cleanliness is also important for maintaining the udder health in the herd [13,14], since failures in farm hygiene resulting in dirty udders pose a risk of mastitis [13,14,15,16,17]. In addition, if the coat of the animal is dirty and wet it will lose some of its insulating capacity, leading to animals having difficulties with thermoregulation [18]. Hence, more energy is needed to maintain thermal comfort and the animals need more feed in order to maintain growth and production rates. In addition to poor welfare due to the risk of health issues or thermoregulation difficulties, dirtiness per se is a severe welfare problem, since urine and manure can cause painful skin burns and dermatitis [19]. Barrientos and co-workers [20] showed that cows in dirty stalls were more likely to suffer from hock injuries, which can result in lameness [21]. Therefore, it is important to not underestimate the impact of farm hygiene and animal cleanliness on animal welfare. Dirty animals at slaughter can also compromise hygiene in the abattoir [5]. According to Hauge and co-workers [22], it is generally accepted that it is more difficult to avoid carcass contamination and preserve good hygiene when dirty cattle enter the slaughter facility.

The causes of dirtiness in cattle are probably multifactorial and several risk factors may be involved, e.g., type of flooring/lying substrate [23,24,25], husbandry system and interior design [3,26], temperature and humidity [5], stocking density [27], parity and outdoor access [28], and deficiencies in hygiene management [5]. These risk factors can be partly eliminated by management actions and good routines by the farmer; for instance, it has been shown that stockperson attitudes can affect animal handling in general, and very likely also the handling of cattle [29,30]. However, while dirtiness in cattle has been proven to have negative consequences for farmers and for cattle, few efforts have been made to rectify the situation. As already described, dirty cattle have been the most common form of non-compliance with Swedish animal welfare legislation for years [1], and are a common welfare issue in other countries, so it is surprising that little seem to be known about prevention. 

The central competent authority responsible for animal welfare in Sweden is the Swedish Board of Agriculture (SBA), while the 21 regional County Administration Boards (CABs) are responsible for carrying out official animal welfare control. The CABs have a joint reference group aiming at improving animal welfare control in Sweden, and this reference group recently initiated a project to deal with the high numbers of dirty cattle observed in official animal welfare inspections. This was done at the request of the Animal Welfare Council, which is made up of representatives from the SBA, the CABs, and the Swedish Food Agency, and which has the task of making animal welfare control more legally secure, equitable, and effective. The project investigated the causes of dirtiness in cattle, with the aim of formulating strategies for future prevention, i.e., improving the compliance rate among farmers. A second aim was to identify why inspectors categorized some farms as compliant even though individual animals were assessed as being ‘not clean enough’.

### Aim

The objectives of this paper are to examine the occurrence of dirty cattle on Swedish cattle farms and compliance with Swedish animal welfare legislation recorded in official animal welfare inspections during the described CAB project, and to assess farmers’ opinions on the reasons for dirtiness and key factors in keeping their animals clean. 

## 2. Material and Methods

Inspectors from 20 of the 21 Swedish CABs participated in the project. A total of 371 farms with dairy and/or beef cattle were inspected within the regular inspection activities of these CABs during two weeks in January/February 2020. Hence, farms were not selected specially for investigating purposes, but would have been inspected during the CABs regular activities regardless of this project. During inspections, particular focus was placed on animal cleanliness. The inspector recorded the number of individual cattle considered to be dirty among the total number of animals on the farm, which is currently not included in regular inspection protocols. To support their on-farm assessments, the inspectors used an inspection protocol and control guidelines issued by the SBA. These documents are already used during animal welfare inspections in Sweden and were not developed specifically for the project. The inspection protocol consists of a checklist where every legal demand is listed in different control points. For each control point, the inspector ticks if the farm is compliant or non-compliant, or if this control point was not assessed or is not applicable on this farm. The control guidelines consist of a written document that describes how an inspector should reason around and assess different requirements and situations [31]. For example, different kinds of dirtiness have different levels of severity, with e.g., mud being categorized as a less severe form of dirtiness than manure and urine dried in the coat. The guidelines also suggest different scales for rating the extent to which the body is dirty, but do not specify how these scales should be interpreted in relation to the legislation. The definition of ‘clean enough’ according to the guidelines is that “animals that are managed in such a way that manure, urine, and sometimes mud does not stick and dry on their bodies”. In the CAB project, the SBA complemented the ordinary guidelines with an additional written guideline stating that “an animal shall be assessed as ‘not clean enough’ if that animal is exposed to, or risks being exposed to, suffering, sickness or abnormal behaviours” [32].

In addition to completing the ordinary inspection protocol, the CAB inspectors were also asked to fill in a questionnaire with questions relating to why the animals were clean or dirty (see translated version in the Appendix A). These questions were answered by the inspectors in dialogue with the farmer. On farms where ‘not clean enough’ animals were found, the farmers were also asked about what could be done to get the animals clean and what they would need to do to prevent dirty cattle. On compliant farms with no dirty animals, the farmers were instead asked about factors they believed to be important for successfully keeping their animals clean. If inspectors categorized a farm as compliant even though dirty animals were observed during inspection, they were asked to state the reason for that categorization. For each question, a set of possible answers was provided and a box for free text was available when the options were not sufficient to describe the inspector’s or farmer’s opinion. The inspectors were not asked to record type of housing system, category of animal (except beef or milk), outdoor access, type of main forage, or type of bedding material. 

The data and information written in the questionnaire was then transferred from the paper sheets to Excel 2016. For descriptive statistical analyses Stata/IC 14.2 (StatCorp, College Station, TX, USA) was used. 

## 3. Results

### 3.1. Types of Farms and Inspections

In total, 371 farms were inspected, out of 14,798 Swedish cattle farms in total in the 20 regions [33]. Of these, 240 were dairy farms and 117 had beef cattle (Table 1). Nine farms had both dairy and beef cattle, while for the remaining five farms the type of production was not stated. The median herd size was 72, ranging from 2 to 2800 heads. Twenty-five percent of the inspections were performed in herds with ≤30 animals (*n* = 93). 

The majority of the inspections (*n* = 265; 76%) were regular animal welfare inspections, while the next largest group was follow-ups on previous inspections (*n* = 67, 19%). Only 13 inspections (4%) were initiated due to notifications of suspected mistreatment of animals. 

### 3.2. Number of Dirty Animals and Legal Compliance on Animal Cleanliness

For 51% of farms inspected (*n* = 188), all animals were assessed as clean enough at the time of inspection (Table 1). For the remaining farms (*n* = 183, 49%), from one to 280 animals were assessed as not clean enough by the inspectors. The majority of these farms had ≤10 dirty animals (*n* = 122). However, in the inspection protocols completed by the inspectors, the cleanliness of the animals was assessed as compliant with Swedish animal welfare legislation on 248 farms (67%) and as non-compliant on 120 farms (33%). Thus, on 59 occasions (16%), the animal welfare inspector assessed the farm as compliant, although ‘not clean enough’ animals were observed during inspection. In three cases, information on compliance with cleanliness was missing. 

The most common reasons stated by the inspectors for categorizing a farm as compliant, despite having ‘not clean enough’ animals present, were: ‘few animals not clean enough’ (*n* = 43, 75%), followed by ‘minor non-compliance’ (*n* = 18, 32%) and ‘temporary non-compliance’ (*n* = 13, 23%) (Figure 1). On 26 occasions (46%), two or more of these reasons for compliance were stated. Information about the reason for compliance was missing for three farms, leaving 56 answers in total. 

On farms where ‘few animals not clean enough’ was mentioned as a reason for compliance, up to 15 animals that were not clean enough were found in the inspection. It was more common for a farm to be assessed as compliant if three or fewer animals were affected, but there were also occasions when a single dirty animal caused the farm to be assessed as non-compliant with the Swedish legislation (Figure 2). 

### 3.3. Reasons for Having Dirty Animals in the Herd

According to the inspectors and farmers, ‘insufficient coat trimming routines’ (*n* = 69, 38%) and ‘absence of routines for grooming dirty animals’ (*n* = 66, 36%), followed by ‘lack of bedding material’ (*n* = 52, 31%) and ‘insufficient cleaning of lying areas or walking alleys’ (*n* = 55, 30%) were the most common reasons for ‘not clean enough animals’ being found during inspections (Figure 3). In most cases, several likely reasons for insufficient cleanliness were stated. 

The behavior of individual animals in choosing a suboptimal place for lying was often given as an explanation when ‘other’ was recorded in the questionnaire (*n* = 20). Bad weather conditions were also a common ‘other’ reason mentioned, especially when animals were housed mainly outdoors (*n* = 14). A third group of explanations identified had to do with feeding (*n* = 7), for instance a sudden change in feed causing diarrhea in some animals, making it more difficult to keep the animals clean. 

### 3.4. Farmers’ Needs and Keys to Success in Keeping Cattle Clean

The responses of farmers with animals categorized as ‘not clean enough’ to the question “*What would you need to do to keep your animals clean?*” and the responses of farmers with only clean animals to the question “*How do you succeed in keeping your animals clean?*” are shown in Figure 4. Several answers could be given. On both types of farms, *management routines* was the factor most commonly mentioned. On farms with clean animals, the trimming and/or grooming of animals was commonly mentioned as an example of a good management routine in keeping animals clean (*n* = 23). The largest discrepancy was found in the answers related to access to bedding material. Only 14% of the farmers with dirty animals (*n* = 26) mentioned ‘better access to bedding material’ as a need, while 63% of the farmers with all clean animals (*n* = 119) mentioned ‘good access to bedding material’ as a key factor in maintaining a good animal hygiene level on their farm. 

Better weather conditions were most often mentioned (*n* = 10) when the option ‘other’ was ticked for farms with dirty animals in the herd. Keeping a low or not too high stocking density was the most commonly stated ‘other’ factor in success for farms with clean animals (*n* = 15). 

## 4. Discussion 

### 4.1. Dirty Cattle Is Common 

This study showed that dirtiness in cattle is still a frequent form of non-compliance with animal welfare on Swedish farms, with no major change compared with previous reports [1]. In this study, the number of farms with dirty animals was almost 50%. However, the data do not show whether there were differences between different categories of animals, e.g., milking cows, dry cows, bulls, heifers, young cattle, and calves, or between housing systems, e.g., tie-stall, loose-housed, or outdoor cattle. Previous studies have shown a higher level of non-compliance [3] or worse welfare [8,34] in tie-stall housing compared with free-stall housing of dairy cows. In Sweden, tie-stall housing is still permitted for some categories of cattle, mainly dairy cows, with the total proportion of cattle kept in this system reported to be lower than 30% [35]. 

As mentioned previously, dirtiness is a welfare issue not only in Sweden, but also in other countries. Most countries do not have specific animal welfare regulations for cattle and do not set specific requirements on the appearance of individual animals, i.e., animal-based measures. This may be one reason why this welfare issue receives so little attention, as no violations are committed if there are no such requirements in the regulations. However, there are countries having national legislation requiring animals to have access to dry and clean lying areas. A resource-based requirement that is closely connected to the cleanliness of animals. However, in the future, the increasing focus and demand for animal-based indicators in both legislation and private assurance schemes [36,37,38] could perhaps draw attention to how animal-based requirements can best be implemented, measured, and enforced. 

Dirty cattle still receive little attention in the industry, despite having an effect on farm profitability. From the perspective of the farmer, the level of knowledge of the individual farmer on the effects of dirty animals may be insufficient. From the perspective of the industry, the issue has not been regarded as important to date, although in 2020, the media in Sweden began drawing attention to dirty cattle on dairy farms, which has created some movement within the industry. It has also been shown that animal welfare issues raised in the media can have an effect on consumer demand [39]. From the perspective of animal welfare inspections and the SBA, the measures taken, and why they are not functioning, can be discussed. Since this form of non-compliance returns year after year, the enforcement of compliance with the legislation needs to be considered. Enforcement work should examine why it is so difficult for cattle farmers to have ‘clean enough’ animals and how better compliance can be achieved. The SBA very recently presented an animal welfare strategy for Sweden where better compliance is one of the main goals, together with an official control that is effective and uniform [40]. 

### 4.2. Challenges in Interpretation and Assessment 

The animal welfare legislation clearly sets a minimum level of animal welfare. Thus, at the time of an animal welfare inspection, animals do not need to be perfectly clean, but ‘clean enough’. Even when cattle in this study were assessed as ‘not clean enough’ during on-farm inspection, the inspector did not always categorize the farm as non-compliant with animal welfare legislation. On the contrary, on multiple occasions the inspector made the assessment that, despite not having clean enough animals, the farm was in compliance with the legislation since the herd as a whole was assessed as ‘clean enough’. The most common comment made in relation to this assessment was that ‘*only a few animals*’ were too dirty. Assessing only a proportion of animals in a herd is becoming more common in animal welfare assessments, e.g., in the Welfare Quality^®^ system [41]. This way of measuring animal welfare is useful for detecting systematic problems at farm level but is in contravention of animal welfare legislation in Sweden, which does not focus on measuring overall animal welfare level at a farm, but on protecting the individual animal [42]. This is also clearly apparent from the SBA control guidelines, which state that “*Suitable measures must be taken if one animal is affected, since the legislation is written from the perspective of an individual animal*” [31]. Hence, if only one animal is not clean enough, this is clearly a case of non-compliance and measures must be taken by both the inspector and the farmer. 

This study revealed that animal welfare assessments can be rather subjective. This indicates a need for better guidance from the SBA on animal welfare inspections, in order to achieve high inter-observer agreement. This is in line with findings in France [43] and Denmark [44]. Use of vague wordings, such as ‘clean enough’, in regulations is quite common [11,45], and increases the risk of subjective assessments [45,46,47]. Further, compared with resource- and management-based requirements, animal-based requirements are more time-consuming to measure [48] and more difficult to assess in a uniform way [49], and hence require much training to perform [50,51]. However, animal-based requirements and indicators are important in coming closer to measuring the actual welfare status of the individual animal, and not only considering risk factors for welfare through resource- and management-based requirements [51,52]. Use of animal-based indicators is one way to evaluate whether animal housing and management are functioning as the legislation intends. The European Commission is considering increased use of animal-based measures in EU legislation [36]. Animal-based requirements are valuable and important but need special attention when assessing compliance. The written SBA guidelines currently used in Sweden do not seem to provide adequate support for animal welfare inspectors or are perhaps not fully accepted by all inspectors. If inspectors question the methods they are required to use, some might be expected not to strictly follow the guidelines [43]. Therefore, new strategies and methods need to be developed and implemented in order to achieve more uniform assessment of compliance and non-compliance in relation to the cleanliness of cattle. 

Apart from official animal welfare inspectors, farmers also need to understand how to interpret ‘clean enough’ and when ‘not clean enough animals’ are considered to be in compliance or non-compliant with the legislation. In the free text space in the questionnaire, one of the inspectors commented ‘*Farmer disagrees on animals being dirty*’, showing that the interpretation of cleanliness is not always straightforward. Such statement also raises questions concerning why some farmers does not see dirtiness as a problem. Is it due to lack of education or a case of “home blindness”? In addition, professional farm advisors and veterinarians need to know the legally acceptable level of cleanliness in cattle. Staaf Larsson and co-workers [53] found that professionals with different backgrounds and experiences viewed and assessed dirtiness in cattle differently. Further, with the use of private standards and quality assurance schemes increasing in society, it is important to include these actors in the discussion, since private standards are commonly based at least partly on the requirements set in national legislation [11,54]. There will also be others making assessments on what is ‘clean enough’ and when a farm is compliant or not. Previous studies have shown that even when a requirement in a private standard is expressed in the same way as in national legislation, the way of measuring may be quite different [3,11]. According to Lundmark and co-workers [55], policymakers need to consider the whole animal welfare inspection arena, and not simply their own regulation or assurance scheme as an isolated entity.

### 4.3. Actions and Attitudes

There is a risk that the uncertainty in how ‘clean enough’ is assessed will result in passive farmers. In this study, farmers with dirty cattle cited different management routines, such as insufficient coat trimming and grooming, insufficient cleaning of lying areas or walking alleys, and lack of bedding material as the most common reasons for dirty cattle. Farmers with ‘clean enough’ animals believed to a higher degree that management routines, sufficient staffing levels, bedding material, and good physical/mental health are key factors in keeping clean animals. Farmers with ‘not clean enough animals’ believed that shorter queues to slaughter would solve the problem of dirty animals to a higher degree. 

Dirtiness in cattle is a multifaceted problem according to the literature, and previous studies have identified risk factors in housing and interior design that contribute to hygiene problems [5,23,26]. Interestingly, and somewhat surprisingly, in this study housing conditions were not mentioned as often as various management routines when farmers were asked about the reasons for having clean or dirty animals. Previous studies have concluded that cattle need enough resting time for prevention of lameness and hock lesions [21,56,57], indicating that the issue of dirty cattle should be taken very seriously. Dirty cattle could be an indicator of animals not having a suitable dry bedding area, which can result in animal welfare-related skin lesion issues, and higher prevalence of lameness and hock lesions. Thus, dirtiness in cattle is a negative economic factor for farmers and can also be an indicator of poor overall herd health.

In this study, several farmers blamed dirtiness on the behavior of individual animals in choosing a suboptimal place for lying. If cattle are free to choose, they will choose a lying area that is soft [58,59], dry and clean [60,61,62], and has a lot of bedding material [63]. Since lying behavior in cattle is strongly linked to comfort and welfare [64,65,66], it is unlikely that cattle will voluntarily choose to lie down somewhere wet and dirty. Therefore, the reasons behind animal behavior in this regard must be evaluated and suitable measures taken by the farmer.

Finally, there is a possible interaction between the farmer’s attitude to animals and the handling regime. Several studies have shown that attitudes to animals influence the way in which animals are handled [30]. Hemsworth and colleagues [30] showed that attitudes, and handling, can be improved by providing species-specific education on animal care. Glanville and co-workers [67] suggested that in efforts to achieve a change in attitudes, one important component must be to inform farmers about the consequences of their handling and about the impacts of a change in handling regime on the animals, but also on the farmer’s situation. It is possible that the issue of dirtiness in cattle can be prevented through suitable education and training. 

## 5. Conclusions

This study showed that cattle in Swedish dairy and beef production are currently not clean enough, and dirty cattle was present at 49% of the farms Questionnaire responses clearly indicated that inspectors and farmers both believe that cleanliness in cattle is primarily connected to routines on the farm, e.g., trimming and grooming of animals. Both the farmers with clean cattle and farmers with cattle not clean enough thought management routines was the main key factor to keep animals clean. This is a promising result from an animal welfare point of view since management routines and strategies can be improved with the right measures. However, there was also some discrepancies between farmers’ views. Farmers with clean animals saw good access to bedding material as one of the keys to success for having clean animals, while few farmers with not clean enough animals did see better access to bedding material as a key factor to improve cleanliness.

Despite dirty animals being present, animal welfare inspectors did not always categorize the farm as non-compliant. We conclude that a better official guidance for inspectors and farmers is needed, in order to support their assessments of when dirtiness in cattle represents non-compliance with animal welfare legislation. 

## Figures and Tables

**Figure 1 animals-11-00945-f001:**
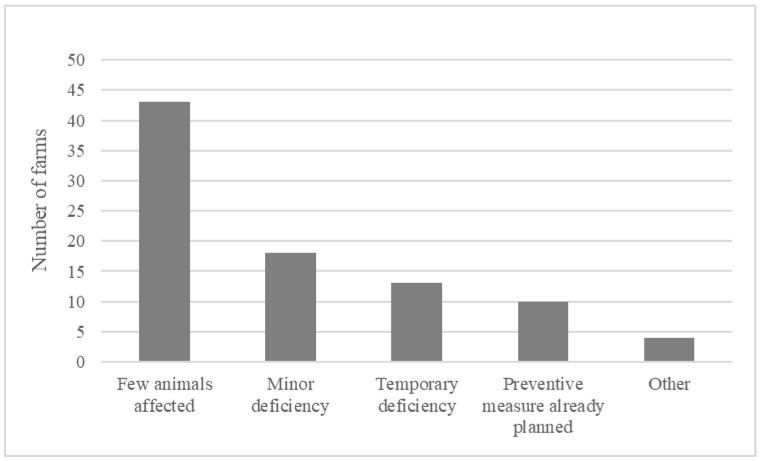
Reasons stated by animal welfare inspectors for categorizing 56 farms as compliant with Swedish animal welfare legislation on animal cleanliness, despite dirty animals being observed during inspections. One or several reasons could be given.

**Figure 2 animals-11-00945-f002:**
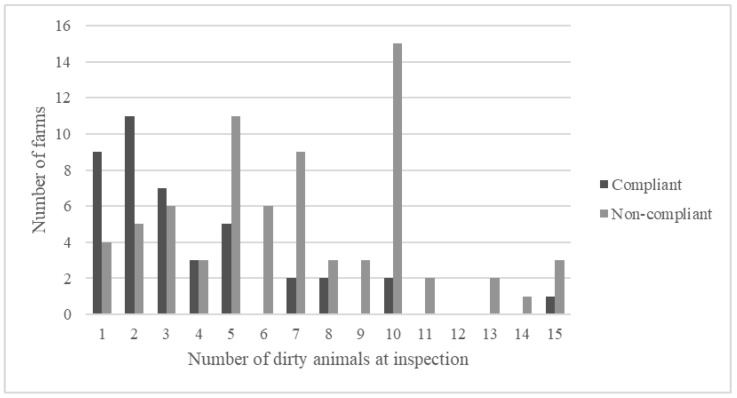
Assessments of compliance with Swedish animal welfare legislation on animal cleanliness during animal welfare inspections on 115 farms where up to 15 animals were found to be not clean enough.

**Figure 3 animals-11-00945-f003:**
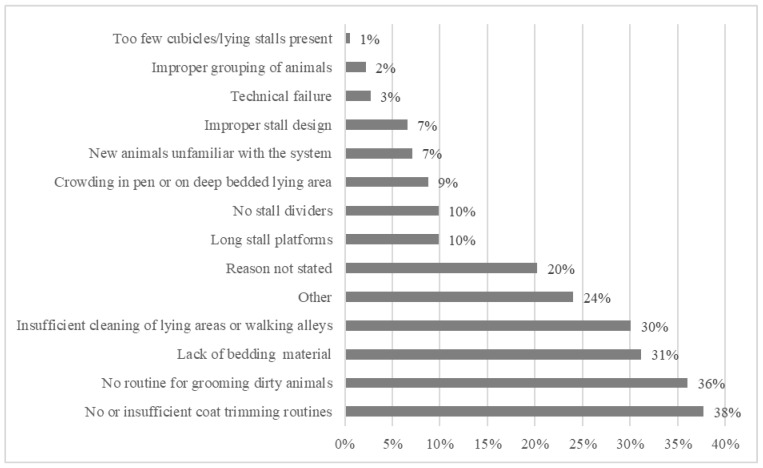
Reasons given for why few or several animals in the herd were dirty during animal welfare inspections on 183 farms. More than one reason could be given.

**Figure 4 animals-11-00945-f004:**
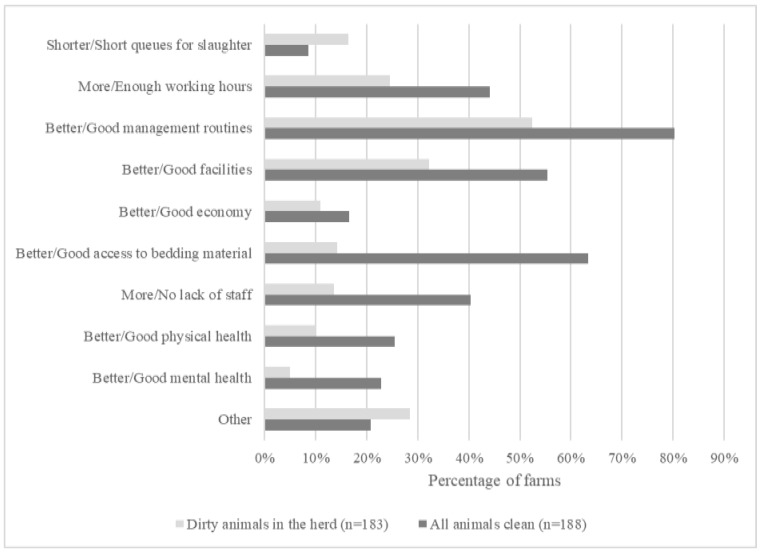
Responses by farmers with dirty animals in their herd on what they would need to keep their animals clean, and by farmers with no dirty animals in their herd on how they succeed in keeping all their animals clean. More than one reason could be given.

**Table 1 animals-11-00945-t001:** Information about the 371 Swedish cattle farms visited for animal welfare inspections during Jan–Feb 2020.

		Farms with Clean Animals Only (*n* = 188)	Farms with Animals Not Clean Enough Present (*n* = 183)
Type of production, number of farms	Beef	135	105
	Dairy	49	68
	Beef & Dairy	2	7
	Missing	2	3

Type of inspection, number of farms	Regular	146	119
	Follow-up	24	43
	Suspected mistreatment	6	7
	Notification from abattoir	2	0
	Missing	10	14

Herd size, number of animals ^1^	Median	56	85
	Min–max	2–768	2–2800
	25th–75th percentiles	20–131	43–186

Number of animals not clean enough at inspection ^2^	Median	-	7
	Min–max	-	1–280
	25th–75th percentiles	-	3–18

^1^ Herd size data missing for two farms. ^2^ Data on number of dirty animals at inspection missing for three farms.

## Data Availability

Restrictions apply to the availability of these data. Data was obtained from 20 Swedish County Administrative Boards and are available from the authors with the permission of the individual County Administrative Board.

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
