# Peer review of "Cattle Cleanliness from the View of Swedish Farmers and Official Animal Welfare Inspectors"

_animals, 2021, doi:10.3390/ani11040945_

Round 1

Reviewer 1 Report

My comments is uploaded in ta file

Author Response

Reviewer (reviewer 1 had several inputs on the same topic about clarifying the questionnaire and who´s view we are presenting):

The paper has a lot of interesting points but there is a major issue around the questionnaire – did both farmers and inspectors fill out the same scheme – and why do you in the aim refer to that this is to study the farmers opinions when you later write “in the inspectors and farmers opinions..”. I have asked question about this in Material and Methods and in the results.

I think it is important for us as readers to know whether you have asked both groups in the same questionnaire or if.

Abstract, Line 29

Uou write abut the aim that it was .. among others “.. to study the farmer’s opinion on

reasons for failure and keys to success.”

Line 36

– In the inspectors and farmers views… But you do not study the inspectors view, do you ?

We know what they report to the authorities –but their views ?? In your aim , line 97, it is

made clear that the aim is to study the farmers opinions or reasons ….

So including the inspectors views here seems a wrong statement. The occurrence of dirty

cattle is studied but not inspectors views…

Line 36

– In the inspectors and farmers views… But you do not study the inspectors view, do you ?

We know what they report to the authorities –but their views ?? In your aim , line 97, it is

made clear that the aim is to study the farmers opinions or reasons ….

So including the inspectors views here seems a wrong statement. The occurrence of dirty

cattle is studied but not inspectors views…

Material and methods. Line 117-125

Firstly - a reference or link to the questionnaire is missing – It should be offered as

supplementary material in English.

Second – I read it as if the questionnaire is for the farmer (and the inspector fill it out) but it

also seems that the inspector could write his /hers own comments ?? (line 124, … to describe

the inspectors or farmer’s opinion. So inspectors could also write free text, but could they

also in other places in the questionnaire write their own opinions. If we have two different

schmes or questionnaires they should both be offered as supplementary material.

It is confusing if inspectors and farmers filled out the same questionnaire or had each a

column ? Or each a questionnaire? This should be made clearer. I would expect that what

you had from the inspectors are mainly their protocols, right ? Not comments in a

questionnaire or …. ? it affects very much how I read the results, it seems a bit of a mess and

must be made more clear.

Results, Line 162

“in the inspectors and farmers opinions..” How can they have the same opinions…?

This refers again to my lack of understanding whether the farmers and inspectors filled out

the same questionnaire ? Here they both have opinions. How can it be BOTH the inspectors

and the farmers opinions?

As I understand it the inspectors brought a questionnaire and filled it out on behalf of the

farmers , by asking them during the inspection.

The farmers had several answers to choose between – but it cannot at the same time also be

the opinions of the inspectors. Delete inspectors here or unfold the methods better.

Our response and actions: According to the reviewers comments above we needed to clarify who’s views we were presenting in the text. Therefore, we have clarified the material and methods section. One important clarification is that the first question related to why the animals were dirty was answered after a dialogue between the inspector and farmer. That is why we sometimes says “according to the inspectors and farmers…” in the result section.
We have also added the questionnaire as supplement material to make it easier for the reader to understand the questionnaire and method used.

Reviewer: Line 33

… filled in a questionnaire when asked the farmers questions .. – is that correct English? A questionnaire which asked… sound more correct.. or THAT asked..

I am not a native English speaker so I will just comment on the language if it really sounds wrong in my ears..

Our response and actions: The manuscript have now been checked by a professional English speaking person, and we have modified the language according to this persons suggestions.

Reviewer: Also, did you use any literature to develop the questionnaire or to analyze it ? I lack

references in case you were inspired using literature and I think we need that in general when

it comes to methodology that does not just “come out of the blue”.

Our response and actions: The questionnaire was designed by the County Administrative Board based on their experience during controls and what they believed was possible to conduct on a regular inspection without being too time consuming. Hence, the questionnaire did not “just come out of the blue”, but it was not developed from scientific literature studies.

Reviewer: Line 200

Will it not end up that the tie stalls will be phased out? In case yes, when ?

Our response and actions: There are no deadline for tie stalls in Sweden. At least not yet. Hence, we cannot write that.

Reviewer: Line 2005.

An interesting start in the discussion - but is it true that the problem cannot be seen in other

countries because they are not using animal based measures in their national legislation?

I know for sure that it is a big problem in Denmark too, also seen in the reports from the

authorities though the national Danish legislation for cattle does not talk about dirty animals

but talk about that animals must have access to a dry lying area.

It is also a very close connection to animal welfare to have access to a dry area to lie down.

And it is often seen in Danish animal welfare inspections that cattle are prevented to lie down

because they have a dirty lying area (and in the same connection it can be seen as lack of

cleanliness)

Our response and actions: We have made some changes in the discussion to mention the relation between dirty/wet lying areas and dirty animals.

Reviewer: 253: I wish you would be a bit more precise here and specific with suggestions– try to

suggest what you mean by new strategies ? You mention earlier training of inspectors –

could you specify this ? Is this what is needed and in what areas do they need training?

Our response and actions: From the results of this study, we have identified the need for better training of inspectors and strategies for more uniform assessments to be developed by the Board of Agriculture. However, in this paper we cannot draw any conclusion or reason about how this training and strategies should be designed.

Reviewer: 260: very interting. I have heard the same notion from inspectors in my country – that the

farmers did not SEE the dirt or disagreed about dirt being a problem. Could be unfolded a bit

more. Is it a part of the problem that the farmers have lack of education or “stable-blindness”

or simply that they do not regard dirt as a problem, and why not ? Here a more qualitative

approach is needed. Questionnaires can give a lot of answers but also rise a lot of questions

and very seldom bring us into a closer understanding of “why”.

Our response and actions: This is a good point. Why does some farmers disagree about dirty cattle being a problem? We have added some more reasoning in relation to this question. However, since this was only one comment we will not draw any strong conclusions from that. We have mentioned earlier in the discussion that farmer knowledge concerning dirtiness may be insufficient.

Reviewer: Line 166 animals choosing a suboptimal place for lying --- is an interesting way of

formulating a problem that somehow must connect to other problems (too small boxes, lying

areas not dry..animal behaviour ) and also a problem about animals who do not lie down

enough because the places to lie are sub-optimal. Hope this will be somehow discussed.

The same is the interesting lines from 290 to 294 about farmers opinions that the animals

choose a suboptimal place – showing that farmers here seem to know very little about

animals behavioural needs . Could be an action or suggestion from you to look into how

farmers are educated today in Sweeden and if there is any connection here to lack of

knowledge and the blindness that some farmers show. You do line up to that area in line 299-

302 so I am happy to see that.

Our response and actions: We agree, we also find these points to be interesting. Hence, this is discussed under section 4.3 Actions and attitudes.

Reviewer 2 Report

Accepted for publication 

Author Response

The manuscript have now been checked by a professional English speaking person, and we have modified the language according to this persons suggestions. 

We have also modified the material and method section and made the conclusion more solid.

Reviewer 3 Report

I consider that the evaluated paper is interesting, well presented, although, with the available data, the authors could have reached more solid conclusiones

Author Response

Reviewer: The authors could have reached more solid conclusions.

Our response and actions: We have rewritten the conclusion a bit to make it more solid and to better summarize the findings.

Reviewer 4 Report

I think that the subject of the work is of interest and that the topic of the manuscript is appropriate for the Journal. The information is of significant interest to the Journal's readers. However I suggest some correction throughout the manuscript. In particular, I think that introduction could be improved by better stressing the concept of welfare in farming animals  emphasizing the significant increase of interest showed by scientific community on this field to enhance animal health status and welfare. Moreover, English language should be improved throughout the manuscript. The conclusion section should be rewritten in order to better summarize the findings and emphasize the significance of the study. In view of such consideration, I suggest that the study could be suitable for publication after minor revision.

Specific Comments

The title accurately reflects the major findings of the work. The abstract adequately summarize results and significance of the study. However, methodology description need to be improved. Authors should better indicate the sampling procedure as well as the statistical analysis applied on the obtained data.

Keywords represent the article adequately.

The introduction section is well written and it falls within the topic of the study, and Authors cited appropriately bibliographic information. I think that Authors should improve this section by better stressing the concept of welfare in farming animals emphasizing the significant increase of interest showed by scientific community on this field to enhance animal health status and welfare. At this regard, at the beginning of the section, I suggest to add the following information “As for other farmed animals, health and biological functioning of cattle are often prioritized. Despite the homeostatic mechanisms to maintain blood parameters within physiologic levels including changes in metabolites and hormones during peculiar life periods of farmed animals are well studied (Fiore E., et al., PLoS One, 2018, 13(4): e0193803; Piccione et al., Journal of Dairy Research, 2011, 78: 421-425; Arfuso et al., 2016, Archives Animal Breeding, 95, 429-434; Fiore et al., Animal Production Science, 2017, 57 (6): 1007-1013), the impact of farm hygiene on animal welfare is often underestimated. In view of the general concepts of animal welfare involving adaptations of normal physiology and behaviour loading to health status that ultimately increases productivity, and considering that external factors including food availability, environmental and management conditions could influence the behavior of animals, the improvement of management conditions including good practice of cleanliness, trimming and/or grooming of farmed animals  become key factor to improve the health status and welfare of animals as well as to enhance productivity in livestock (Giannetto C. et al., Journal of Veterinary Behavior, 2018, 23: 97-100; Fazio F. et al., Journal of Veterinary Behavior, 2018, 26: 5-10)."

The section of Materials and Methods is clear for the reader and it meticulously describes the methods applied in the study. However, Authors should check this section and correct many punctuation errors and some mistakes.

Results section as well as Discussion section is clear and well written. The findings obtained in the study were well discussed and justified with appropriate references.

Authors should rewrite the conclusion section in order to better summarize the results and the significance of the study.

Tables and figures are generally good and well represent the results of the study.

Authors should check and standardize the references in the list according to journal guidelines.

Author Response

Reviewer 4

Reviewer: I think that the subject of the work is of interest and that the topic of the manuscript is appropriate for the Journal. The information is of significant interest to the Journal's readers. However I suggest some correction throughout the manuscript. In particular, I think that introduction could be improved by better stressing the concept of welfare in farming animals emphasizing the significant increase of interest showed by scientific community on this field to enhance animal health status and welfare.
The introduction section is well written and it falls within the topic of the study, and Authors cited appropriately bibliographic information. I think that Authors should improve this section by better stressing the concept of welfare in farming animals emphasizing the significant increase of interest showed by scientific community on this field to enhance animal health status and welfare. At this regard, at the beginning of the section, I suggest to add the following information “As for other farmed animals, health and biological functioning of cattle are often prioritized. Despite the homeostatic mechanisms to maintain blood parameters within physiologic levels including changes in metabolites and hormones during peculiar life periods of farmed animals are well studied (Fiore E., et al., PLoS One, 2018, 13(4): e0193803; Piccione et al., Journal of Dairy Research, 2011, 78: 421-425; Arfuso et al., 2016, Archives Animal Breeding, 95, 429-434; Fiore et al., Animal Production Science, 2017, 57 (6): 1007-1013), the impact of farm hygiene on animal welfare is often underestimated. In view of the general concepts of animal welfare involving adaptations of normal physiology and behaviour loading to health status that ultimately increases productivity, and considering that external factors including food availability, environmental and management conditions could influence the behavior of animals, the improvement of management conditions including good practice of cleanliness, trimming and/or grooming of farmed animals become key factor to improve the health status and welfare of animals as well as to enhance productivity in livestock (Giannetto C. et al., Journal of Veterinary Behavior, 2018, 23: 97-100; Fazio F. et al., Journal of Veterinary Behavior, 2018, 26: 5-10)."

Our response and actions: We agree that it is important to not underestimate the impact of farm hygiene on animal welfare. However, we believe that we have made this quite clear both in the introduction and in the discussion. The suggested new text from the reviewer does not add much new or significant information. However, we have clarified in one sentence the importance of farm hygiene on animal welfare, Line 70-71.

Reviewer: English language should be improved throughout the manuscript.
The section of Materials and Methods is clear for the reader and it meticulously describes the methods applied in the study. However, Authors should check this section and correct many punctuation errors and some mistakes.
Our response and actions: The manuscript have now been checked by a professional English speaking person, and we have modified the language according to this persons suggestions.

Reviewer: The conclusion section should be rewritten in order to better summarize the findings and emphasize the significance of the study.

Our response and actions: We have rewritten the conclusion a bit to make it more solid and to better summarize the findings.

Reviewer: Methodology description need to be improved. Authors should better indicate the sampling procedure as well as the statistical analysis applied on the obtained data.

Our response and actions: We have clarified what we mean by farms being visited within regular inspection activities of the CABs (Line 106-107). We have also clarified that the analysis are descriptive and that Excel and Stata have been used (Line 140-142).

Reviewer: Authors should check and standardize the references in the list according to journal guidelines.

Our response and actions: We have followed the ”instruction for authors”. If the editor find any mistakes in the handling of references, please let us know.